# MIASurviveMTP: Machine learning for immediate assessment and survival prediction after massive transfusion protocol

Michael D. Cobler-Lichter[ID]*, Jessica M. Delamater, Brianna L. Collie, Nicole B. Lyons[ID], Luciana Tito Bustillos, Nicholas Namias, Brandon M. Parker, Jonathan P. Meizoso, Kenneth G. Proctor

Divisions of Trauma, Surgical Critical Care & Burns, Daughtry Family Department of Surgery, University of Miami Miller School of Medicine and Jackson Memorial Hospital Ryder Trauma Center, Miami, Florida, United States of America

* mdc232@miami.edu

## Abstract

Early triage of trauma patients requiring massive transfusion (MT) may help to marshal appropriate resources and improve treatment and outcome. Artificial intelligence (AI) and machine learning (ML) offer theoretical advantages compared to conventional prediction algorithms but have not been thoroughly evaluated in this population. We hypothesized that AI/ML techniques incorporating all available data in a patient's medical record could achieve similar, if not higher, performance in the prediction of mortality in MT patients as compared to existing models. Patients from the American College of Surgeons Trauma Quality Improvement Project database (TQIP) were retrospectively reviewed. Those receiving ≥ 5 units of red blood cells and/or whole blood within the first four hours of arrival were defined as MT patients. Those receiving ≥10 units were identified as ultramassive transfusion (UMT) patients. ML models were created to predict 6-hour mortality using variables available at different time points, including patient arrival. Of 5,481,046 patients in TQIP from 2017 to 2021, 47,744 received MT and 20,337 of these received UMT. Using only variables available on arrival, MT AUROC was 0.901 [95% CI 0.895–0.910] which increased to 0.943 [95% CI 0.938–0.948] with addition of 4-hour variables. For UMT, arrival AUROC was 0.858 [95% CI 0.846–0.872] and increased to 0.922 [95% CI 0.914–0.931] at 4 hours. ML models reliably predict mortality in both MT and UMT patients. These are the only ML models trained on MT and UMT patients. Future work can focus on prospective implementation of these models with potential direct integration into the electronic medical record. Real-time utilization of comprehensive patient data may enhance clinical decision-making regarding which patients should continue receiving massive transfusion, thus optimizing the allocation of this limited resource.

**Data availability statement:** We note that your Data Availability statement states the following: "All code used in development, training, and evaluation of models is provided in the supplementary material. Source data is the property of the Committee on Trauma, American College of Surgeons (NTDB Admission Year 2017-2021, Chicago, IL) but can be provided on request in conjunction with their approval (https://www.facs.org/quality-programs/trauma/quality/national-trauma-data-bank/datasets/) through application on the American College of Surgeons TQIP website (https://www.facs.org/quality-programs/trauma/quality/national-trauma-data-bank/datasets/)." We also note that you state the following: "Data Availability: While all our code is publicly available (https://github.com/mdcoblermd/MTP), the datasets that were used to develop these models are property of the American College of Surgeons Committee on Trauma (NTDB Admission Year 2017-2021, Chicago, IL) and we are not at liberty to provide this data, though it is available to with an application through the American College of Surgeons directly (https://www.facs.org/quality-programs/trauma/quality/national-trauma-data-bank/datasets/)." At this time, please confirm that we may update your Data Availability statement to the following: "All code used in development, training, and evaluation of models is provided in the supplementary material and on Github (https://github.com/mdcoblermd/MTP). Source data is the property of the Committee on Trauma, American College of Surgeons (NTDB Admission Year 2017-2021, Chicago, IL) but can be provided on request in conjunction with their approval (https://www.facs.org/quality-programs/trauma/quality/national-trauma-data-bank/datasets/) through application on the American College of Surgeons TQIP website (https://www.facs.org/quality-programs/trauma/quality/national-trauma-data-bank/datasets/).

**Funding:** The author(s) received no specific funding for this work.

**Competing interests:** I have read the journal's policy and the authors of this manuscript have the following competing interests: JPM: research support and travel/lodging support from CSL Behring and Takeda Pharmaceuticals; grant support from the U.S. Department of Defense/AMRMC (W81XWH-15-9-0001); speaking honoraria from Cerus Corporation and Memorial Healthcare System; scientific

## Introduction

Massive transfusion (MT), defined as transfusion of ≥10 units of Whole Blood (WB) or packed red blood cells (pRBCs) within 24 hours, or 5 units within 4 hours, is required for approximately 3–5% of trauma patients [1]. With approximately 2.6 million trauma admissions per year in the US, there are 100,000 MTs for trauma each year -- 70% of all blood consumed at trauma centers [2,3]. Though standardization of Massive Transfusion Protocols (MTP) has led to reductions in both mortality, postoperative complications, and total volume of product transfused, these patients still face high mortality rates [4–8]. To make matters worse, the number of people donating blood has declined 40% in the last 2 decades [9]. Thus, judicious use of this limited resource is an important consideration, especially in resource-limited settings.

Both the Assessment of Blood Consumption (ABC) score and the Massive Transfusion Score have been applied in mortality prediction after MT but have only achieved 60–70% discrimination by area under the receiver operator curves (AUROC), likely because these scores, along with virtually every other MT-related scoring system, have been designed to predict who will require MTP, not who will survive once it has been initiated [10–12]. In current practice, the decision to initiate an MTP is typically made within minutes of patient arrival, often based on initial vital signs and clinical gestalt, with transfusion beginning almost immediately once activated. However, decisions about continuing transfusion and prognostication beyond the initial phase are more complex and rely on evolving physiologic parameters over the ensuing hours.

Artificial Intelligence (AI) and Machine Learning (ML) are rapidly growing fields that excel in their ability to integrate large amounts of data quickly to make real-time predictions. The ability of these models to use substantially more information in making predictions as compared to standard traditional techniques offers a unique advantage now that virtually all patient records are maintained electronically, making access to this data easy. ML algorithms have been developed for prediction of post-trauma complications such as surgical site infection, venous thromboembolism, and mortality in other populations as well as many other surgical and non-surgical populations [13–15]. These methods have also been refined to predict mortality in Critical Administration Threshold positive (CAT+) patients, those requiring ≥3 units of pRBCs in the first hour. However, the only model reported in the literature is based on < 2,000 CAT+ patients, not true MT patients, and its primary focus was prediction of need for MT [16].

To fill this gap, we aimed to utilize AI and ML techniques to predict survival in trauma patients undergoing MT using a large national dataset. We hypothesized that ML models trained on MT and ultramassive transfusion (UMT) patients could achieve similar, if not higher performance compared to pre-existing models for CAT+ patients, and that these predictions could be improved further with the incorporation of more clinical information that becomes available as the resuscitation progresses. These evidence-based mortality prediction tools have the potential to guide more informed blood allocation, especially in resource-limited settings where transfusion resources may be scarce.

advisory board for Haemonetics, Inc.; editorial board for Trauma Surgery and Acute Care Open and Journal of Surgical Research. NN: NN receives compensation as an advisory board member for Molnlycke, grand funding from Humacyte, and is on the editorial board of Surgical Infections, and the Journal of Trauma and Acute Care Surgery. All other authors: no disclosures. This does not alter our adherence to PLOS ONE policies on sharing data and materials.

## Methods

Per recommendations by the EQUATOR network, the Transparent Reporting of a multivariable prediction model for Individual Prognosis Or Diagnosis (TRIPOD+AI) guidelines were followed, and the checklist is available in S1 Appendix [17]. This study was exempt from institutional review board approval because of use of a de-identified national database.

### Data source

The American College of Surgeons Trauma Quality Improvement Program (TQIP) dataset from 2017–2021 was retrospectively reviewed. Patients who underwent any transfusion were identified using International Statistical Classification of Diseases and Related Health Problems Version 10 (ICD 10) codes. From this cohort, those who underwent MT, defined as ≥5 units of pRBC and/or WB within the first 4 hours of resuscitation, and UMT, defined as ≥10 units of pRBCs and/or WB within the first four hours, were selected. 6-hour mortality was derived by combination the mortality variable with the time to discharge variable (those who died with a time to discharge <=6 hours were identified as mortalities within 6 hours). Those with missing data for mortality were excluded.

### Data preprocessing

Initial exploratory data analysis was performed using IBM SPSS Statistics version 28 (International Business Machines Corp, Armonk, New York). ML model development was done using Google Colab servers running Python 3.10.12 (Python Software Foundation, Beaverton, OR) with following packages: scikit-learn version 1.5.2, tensorflow version 2.9.1, DeepTables version 0.2.6, XGBoost version 2.0.2, SHAP version 0.46.0 [18–23]. All code is available in S2 Appendix.

Of all available variables in TQIP, only those that would have been available on arrival were included in the initial model, termed the arrival model, to simulate variables that would be available upon initiation of MTP. Additional 4-hour models were developed separately that also included variables that would be available after the first 4 hours of resuscitation, such as 4-hour transfusion volume, to refine the initial prediction made by the arrival model. In addition to the standard variables available in TQIP, multiple other variables were derived from available data, including body mass index, total number of patients transfused at a given facility, mortality rate at a given facility, and average volume of blood products per patient administered at a given facility. Patients (n=88) that had facility mortality rates of 0, or total facility deaths of 0 or 1 had their facility-level data censored and treated as missing data to prevent perfect separation of the data based on this single variable. Abbreviated Injury Scale (AIS) codes were processed to derive 32 additional binary variables that correspond to the presence or absence of injury in clinically relevant body regions/organs (S3 Appendix) for inclusion in the 4-Hour models. See S4 Appendix for an exhaustive list of variables contained in each model.

All categorical variables were one-hot encoded (a method of transforming a categorical variable with $n$ levels into $n$ binary variables that are more easily handled by

predictive modeling). Any variable with >50% missing data was excluded. For categorical data with <50% but >0% missing data, missing values were replaced with the most common value across the data set for each variable (mode imputation). For continuous data, the missing values were replaced with the median value of the variable across the dataset (median imputation), a commonly described technique for handling missing data [24]. Random imputation was also tested for key continuous variables, using values drawn from a synthetic distribution matched to the variable's mean and variance, to ensure model performance was not overly dependent on patterns of missingness. To assess the impact of the missingness threshold used for variable exclusion, we conducted sensitivity analyses using five thresholds (10%, 20%, 33%, 50%, and 66%), with empiric selection of the best-performing threshold overall for the final model threshold (see S5 Appendix for sensitivity analysis results). Continuous variables were normalized by subtraction of the mean of each variable and scaling to unit variance. All time-related variables are measured from patient arrival to the trauma bay.

## ML model development

Given the potential for these predictive models to influence decisions on continuation or stoppage of MT in critically ill patients, we chose an early mortality timepoint (6-hour) as our outcome. 6-hour mortality was chosen as the endpoint as early mortality in patients requiring MT is most reflective of hemorrhage-related death, the outcome most directly targeted by MT protocols, and is a commonly used outcome in hemorrhage-control literature. Each later timepoint would have progressively higher mortality rates and would be more influenced by specific in-hospital treatments compared to the initial resuscitation, inflating the potential error in these models and therefore the potential harm.

The models tested included logistic regression (with both LASSO and Ridge regularization), gradient-boosted decision trees (GBDT), random forest, k-nearest neighbor, and artificial neural networks. Each model was optimized against the AUROC. The data were randomly split into training (80%) and testing (20%) sets. The training set was again split into true training (80%) and calibration (20%) sets for model calibration. Grid search with fivefold cross validation was used to determine optimal hyperparameters (tested combinations of hyperparameters and final specifications available in S2 Appendix). Models were calibrated after fitting and hyperparameter optimization.

Two different model set were developed: the MT models that were trained on any patient who received ≥5 units of either pRBC or WB in the first four hours of resuscitation, and the UMT models that were trained on patients who received ≥10 units in the first four hours.

## Model evaluation

The AUROC with 95% confidence intervals (CI) was calculated using bootstrapping methods with 1,000 iterations and served as the primary performance metric for each model. The model with the highest AUROC was chosen as the best performing model for each time interval (arrival and 4-hour) and was compared against a more traditional regression-based approach (LASSO) using paired bootstrap testing with 2,000 iterations. Secondary performance metrics included sensitivity, specificity, positive predictive value, area under the precision recall curve (a measure of a model's ability to correctly identify the positive class across various classification thresholds and that may help inform model performance in cases of imbalanced data with low prevalence rates), F1 score (the harmonic mean of positive predictive value and sensitivity), and the Brier Score (mean squared difference between predicted probabilities and actual outcomes). Secondary metrics that depend on positive classification threshold chosen (sensitivity, specificity, positive predictive value, negative predictive value, and F1 score) were reported at two different classification thresholds: the threshold which maximizes F1 score and the threshold that achieves 90% specificity.

Models were calibrated after fitting. Calibration success was assessed by examining reliability diagrams in which predictions are grouped into deciles of predicted risk and compared the mean predicted probability of each decile to the observed proportion of positive outcomes. Plotting these points against the diagonal (perfect calibration line) indicates whether the model is under- or overestimating risk. Agreement between predicted probabilities and the observed event

rates signifies successful calibration and was therefore examined in accordance with best practice guidelines for development of ML-based clinical prediction models [25]. 95% CI's were generated with bootstrap resampling of the dataset with 1000 iterations.

Similarly, the potential clinical utility of the predictive model was evaluated using decision curve analysis across a range of threshold probabilities. Higher net benefit implies that using the model at that threshold would yield better clinical outcomes than either of two baseline strategies: treating all (assuming everyone died) or treating none (assuming no one died), accounting for the trade-off between true positives and false positives [26,27].

Finally, to improve interpretability and provide insight into model behavior, we conducted a descriptive analysis of feature importance using Shapley additive explanation (SHAP) scores, a game-theoretical approach that estimates the average marginal contribution of each variable across all permutations. While informative, this feature analysis is secondary to the primary goal of the study, which is predictive modeling of 6-hour mortality in patients receiving massive transfusion [23,28].

## Results

### Patient characteristics

Of 5,481,046 patients in TQIP from 2017 to 2021, 333,987 received at least one unit of any blood product. 47,744 patients received MT and of these, 20,337 received UMT. After excluding patients with missing data for 6-hour mortality (n = 1,675, of whom 800 were also in the UMT group), the study populations were comprised of 46,069 in the MT group and 19,537 in the UMT group.

The 46,069 MT patients had a 6-hour mortality rate of 21.9%, increasing to 41.2% at 30 days. Of these patients, 19,537 underwent UMT, with mortality rates at 6-hours and 30 days of 29.9% and 53.7% respectively. Tables 1 and 2 depict descriptive statistics for the MT (with UMT excluded) and UMT cohorts respectively, stratified by outcome.

### ML prediction performance

A GBDT showed the best performance for predicting 6-hour mortality for both the arrival and 4-hour timepoints. In both the MT and UMT models, performance improved at the 4-hour timepoint, from AUROC of 0.901 [95% CI 0.895–0.910] to 0.943 [95% CI 0.938–0.948] and 0.858 [95% CI 0.846–0.872] to 0.922 [95% CI 0.911–0.931] respectively (Fig 1). On paired bootstrap analysis, all GBDT models outperformed LASSO models (all $p < 0.0001$, Table 3).

All models demonstrated satisfactory calibration as depicted by the reliability diagrams in Fig 2. Decision curve analysis showed all models demonstrated net benefit over both "treat none" and "treat all" baseline strategies (Fig 3) for all decision thresholds under 0.95. Secondary performance metrics are displayed in Table 4.

Evaluation with ShAP in all models identified the clinical factors having the strongest impact on model prediction, and were roughly similar between MT and UMT cohorts. On arrival, these were ED Glasgow Coma Scale (GCS), lowest systolic blood pressure, and patient weight/body mass index (BMI). At the four-hour timepoint, these determinant remained similar but total 4-hour transfusion volume became the single most important predictor in the MT group, and cryoprecipitate administration in the UMT group (Figs 4, 5).

## Discussion

To our knowledge, this is the first study to describe the use of AI and ML methods for real-time prediction of 6-hour mortality in patients undergoing MT and UMT. A GBDT performed the best for all models with AUROC of 0.901 [95% CI 0.895–0.910] in the arrival model, increasing to 0.943 [95% CI 0.938–0.948]in the 4-hour model (0.858 [95% CI 0.846–0.872] to 0.922 [95% CI 0.914–0.931] in UMT models). On descriptive feature analysis, the variables with the highest impact on the model's prediction, on average, included ED GCS, lowest systolic blood pressure, patient weight/BMI, total 4-hour transfusion volume, and administration of cryoprecipitate. These results demonstrate that ML models can use variables available

Table 1. Characteristics of patients receiving MT.

| | Total (N = 47,744) | Survived (n = 27,406) | Died (n = 20,338) | P* |
|---|---|---|---|---|
| Age (yr) mean ± SD | 40.0 +/- 18.2 | 38.4 +/- 16.9 | 42.2 +/- 19.7 | <0.001 |
| Male | 78% (37,116/47,563) | 79.5% (21,692/27,298) | 76.1% (15,424/20,265) | <0.001 |
| Mechanism | | | | |
| Blunt | 59.2% (28,015/47,322) | 55.4% (15,036/27,142) | 64.3% (12,979/20,180) | <0.001 |
| Penetrating | 40.5% (19,174/47,322) | 44.3% (12,025/27,142) | 35.4% (7,149/20,180) | |
| Injury Intent | | | | |
| Unintentional | 59.9% (28,412/47,446) | 56.8% (15,464/27,212) | 64.0% (12,948/20,234) | <0.001 |
| Self-Inflicted | 5.3% (2,535/47,446) | 4.7% (1,278/27,212) | 6.2% (1,257/20,234) | |
| Assault | 31.8% (15,070/47,446) | 35.5% (9,663/27,212) | 26.7% (5,407/20,234) | |
| Lowest SBP mean ± SD | 70 +/- 33 | 79 +/- 25 | 57 +/- 38 | <0.001 |
| Heart Rate mean ± SD | 103 +/-40 | 111 +/- 30 | 91 +/- 49 | <0.001 |
| GCS (median, IQR) | 10 [3 –15] | 14 [6 –15] | 3 [3 –9] | <0.001 |
| Race/Ethnicity | | | | |
| % Black | 29.8% (14,205/47,744) | 31.2% (8,560/27,406) | 27.8% (5,645/20,338) | <0.001 |
| % White | 53.4% (25,506/47,744) | 52.9% (14,498/27,406) | 54.1% (11,008/20,338) | |
| % Hispanic | 15.9% (7,116/44,855) | 16.3% (4,266/26,192) | 15.3% (2,850/18,663) | 0.004 |
| ISS (median, IQR) | 27 [18-38] | 25 [17-34] | 30 [22-43] | <0.001 |
| 4-hour RBCs/WB units | 12.5 [6.0-14.0] | 7.9 [6-11.2] | 10.5 [7.0-18.7] | <0.001 |
| Whole Blood use | 17.3% (8,236/47,744) | 16.8% (4,593/27,406) | 17.9% (3,643/20,338) | <0.001 |
| 6-hour mortality | 21.9% (10,106/46,069) | -- | -- | N/A |
| 30-day mortality | 41.2% (18,993/46,069) | -- | -- | N/A |

* p values for means were obtained from Student's t test or Mann-Whitney U test; p values for proportions were obtained from χ2

Values are reported as % (n) for categorical data, mean +/- SD for continuous normally distributed data, median +/- IQR for continuous non-normally distributed data. SD: standard deviation; SBP: systolic blood pressure; GCS; Glasgow Coma Scale; IQR: interquartile range; ISS: injury severity score; RBC: red blood cell; WB: whole blood.

at the time of MTP initiation to accurately predict early mortality. These predictions outperform traditional regression-based approaches (p < 0.0001, Table 3) and can be refined by the inclusion of more data points as the resuscitation progresses. This information can serve as a useful clinical adjunct for clinicians facing difficult decisions about continued use of limited blood products and may be especially helpful in more resource-limited settings, such as smaller trauma centers or in cases of mass casualty incidents. This may also aid in difficult family discussions surrounding prognostication of this very sick cohort.

MTP is not a single event but an ongoing process, often over hours and involving variable transfusion volumes. While some patients may receive only a few units of blood, others can receive more than 100. Throughout this process, clinicians face multiple decision points where reassessment of resuscitation goals, patient trajectory, and futility becomes necessary. Our model is intended to support decision-making throughout the entire resuscitation process by providing real-time mortality prediction once MT has already begun and resuscitation is ongoing. While emergency and trauma physicians already incorporate evolving physiologic data into their decision-making, these decisions often rely heavily on clinical gestalt. The aim of our model is to quantify that expert intuition using data-driven approaches, creating a reproducible, evidence-based tool that mirrors expert reasoning. This is a goal supported by multiple recent studies citing the need to adapt advanced diagnostics, precision medicine, and patient-tailored strategies to inform clinical care in the context of massive transfusion [29,30].

We envision this model functioning as a decision-support tool for physicians during resuscitation, augmenting rather than replacing clinical judgment, particularly when time and resources are limited. This may be especially beneficial in

**Table 2. Characteristics of patients receiving UMT.**

| | Total (N = 20,337) | Survived (n = 9,087) | Died (n = 11,177) | P* |
|---|---|---|---|---|
| Age (yr) mean ± SD | 38.9 +/- 17.6 | 36.4 +/- 15.7 | 40.9 +/- 18.7 | <0.001 |
| Male | 79.6% (16,124/20,264) | 81.8% (7,437/9,087) | 77.7% (8,687/11,177) | <0.001 |
| Mechanism | | | | |
| Blunt | 55.4% (11,175/20,184) | 48.6% (4,398/9,045) | 60.8% (6,777/11,139) | <0.001 |
| Penetrating | 44.4% (8,969/20,184) | 51.1% (4,620/9,045) | 39.0% (4,349/11,139) | |
| Injury Intent | | | | |
| Unintentional | 55.9% (11,306/20,225) | 50.2% (4,551/9,066) | 60.5% (6,755/11,159) | <0.001 |
| Self-Inflicted | 4.7% (960/20,225) | 4.3% (394/9,066) | 5.1% (566/11,159) | |
| Assault | 35.8% (7,244/20,225) | 41.7% (3,782/9,066) | 31.0% (3,462/11,159) | |
| Lowest SBP mean ± SD | 65 +/- 34 | 76 +/- 27 | 57 +/- 37 | <0.001 |
| Heart Rate mean ± SD | 103 +/- 43 | 114 +/- 32 | 93 +/- 49 | <0.001 |
| GCS (median, IQR) | 7 [3 –14] | 14 [4 –15] | 3 [3 –11] | <0.001 |
| Race/Ethnicity | | | | |
| % Black | 32.3% (6,576/20,337) | 34.9% (3,184/9,121) | 30.2% (3,392/11,216) | <0.001 |
| % White | 50.4% (10,241/20,337) | 49.3% (4,501/9,121) | 51.2% (5,740/11,216) | |
| % Hispanic | 16.2% (3,074/19,004) | 16.9% (1,475/8,750) | 15.6% (1,599/10,254) | 0.018 |
| ISS (median, IQR) | 29 [21-41] | 26 [18-36] | 33 [25-43] | <0.001 |
| 4-hour RBCs/WB units | 15.7 [12.0-23.0] | 14.0 [11.2-19.0] | 17.5 [12.7-26.0] | <0.001 |
| Whole Blood use | 19.5% (3,956/20,337) | 19.0% (1,732/9,121) | 19.8% (2,224/11,216) | 0.132 |
| 6-hour mortality | 29.9% (5,833/19,537) | -- | -- | N/A |
| 30-day mortality | 53.7% (10,494/19,537) | -- | -- | N/A |

\* p values for means were obtained from Student's t test or Mann–Whitney U test; p values for proportions were obtained from χ2

Values are reported as % (n) for categorical data, mean +/- SD for continuous normally distributed data, median +/- IQR for continuous non-normally distributed data. SD: standard deviation; SBP: systolic blood pressure; GCS; Glasgow Coma Scale; IQR: interquartile range; ISS: injury severity score; RBC: red blood cell; WB: whole blood.

lower-volume or lower-acuity trauma centers, where providers may not encounter massive transfusion cases frequently and where reliance on gestalt may be more variable. In these settings, a real-time, data-driven prediction model can help standardize decision-making, reduce uncertainty, and support more confident choices in the face of limited clinical exposure. Similarly, in mass casualty events, where clinical bandwidth and blood product availability can be rapidly overwhelmed, such a tool could aid in triaging ongoing transfusion efforts and prioritizing patients most likely to benefit.

Previous studies identifying when MT is futile in trauma patients have focused on identifying a threshold number of units of blood when continued resuscitation is considered "futile," or have identified risk factors for mortality after MT and/or UMT. For instance, Louden et al identified ≥ 16 units in 4 hours as the point at which mortality rates eclipse 50%, labeling this "heroic," and ≥ 36 units as the at which survival rates approach 0%, labeling this "near futile" [31]. Ang et al then went a step further to describe how patient age should influence these thresholds, with increasing patient age resulting in lower transfusion thresholds to maintain the same mortality rates [32]. A recent narrative review of this topic by Kim et al identified nine articles that examined this topic of when ongoing transfusion in trauma patients is futile, focusing on patients in the UMT range [33]. While the authors note that it is difficult, if not impossible, to decide on an absolute value a transfusion cutoff, they do note that "circumstances vary and the decision to continue transfusions should be individualized". This widely shared view underscores the potential utility of ML-based solutions that provide highly individualized risk estimates, especially given the growing consensus that no single transfusion threshold can universally define futility in massive transfusion. Static numerical cutoffs have proven inadequate, and clinical decisions must instead account for patient-specific and situational factors [34–40].

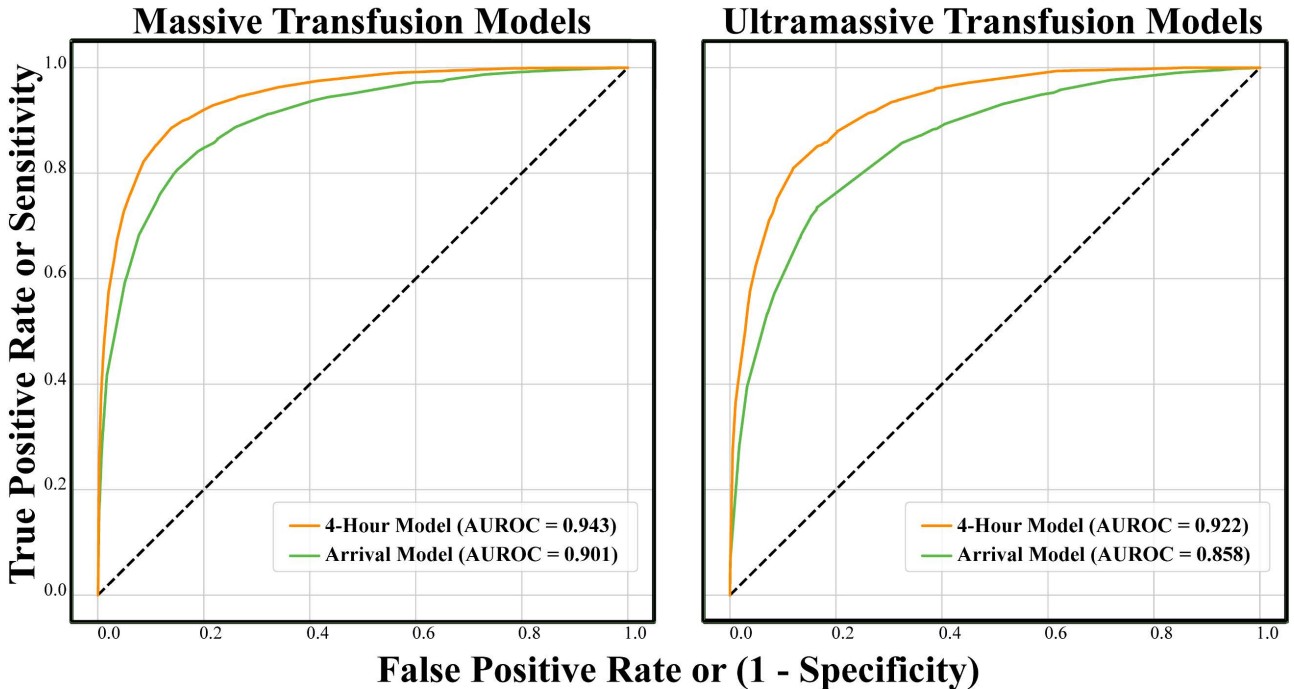

**Fig 1. Receiver-operator curves for prediction of 6-hour mortality in each model.** AUROC: area under the receiver-operator curve.

**Table 3. Comparison of final models to L1 regularization regression models.**

| Model | GBDT AUROC [95% CI] | LASSO AUROC [95% CI] | AUC Difference [95% CI] | p |
|---|---|---|---|---|
| **MT** | | | | |
| Arrival | 0.902 [0.895-0.910] | 0.833 [0.823-0.844] | 0.069 [0.061-0.078] | <0.0001 |
| 4-Hour | 0.943 [0.938-0.948] | 0.904 [0.896-0.911] | 0.040 [0.034-0.044] | <0.0001 |
| **UMT** | | | | |
| Arrival | 0.860 [0.847-0.873] | 0.776 [0.761-0.793] | 0.083 [0.070-0.097] | <0.0001 |
| 4-Hour | 0.923 [0.914-0.932] | 0.881 [0.869-0.893] | 0.042 [0.033-0.050] | <0.0001 |

GBDT: Gradient-boosted decision tree; AUROC; area under the receiver-operator curve; CI: confidence interval; MT: massive transfusion; UMT: ultra-massive transfusion. Point estimates differ slightly in this table as these were obtained via paired bootstrap testing, a slightly different method compared to the rest of the manuscript.

There has been prior work on prediction models in MT patients, but virtually every scoring system and/or algorithm has been developed to predict which patients will require MTP activation in the first place, not who will survive once it has been initiated [11,12,41,42]. Additionally, while there are standardized metrics to quantify injury severity, such as Injury Severity Score and Trauma and Injury Severity Score, these are not available in real time and therefore not useful in mortality prediction or prognostication during the resuscitation. Benjamin et al developed the first ML-based algorithm for mortality risk-estimation in trauma patients requiring transfusions, and was the first proof-of-concept that highly individualized mortality prediction can outperform standard methods of mortality prediction, such as stratifying patients by ABC or Massive Transfusion Scores [16]. While their algorithm was based on a cohort of CAT+ patients, not MT patients, their Tier

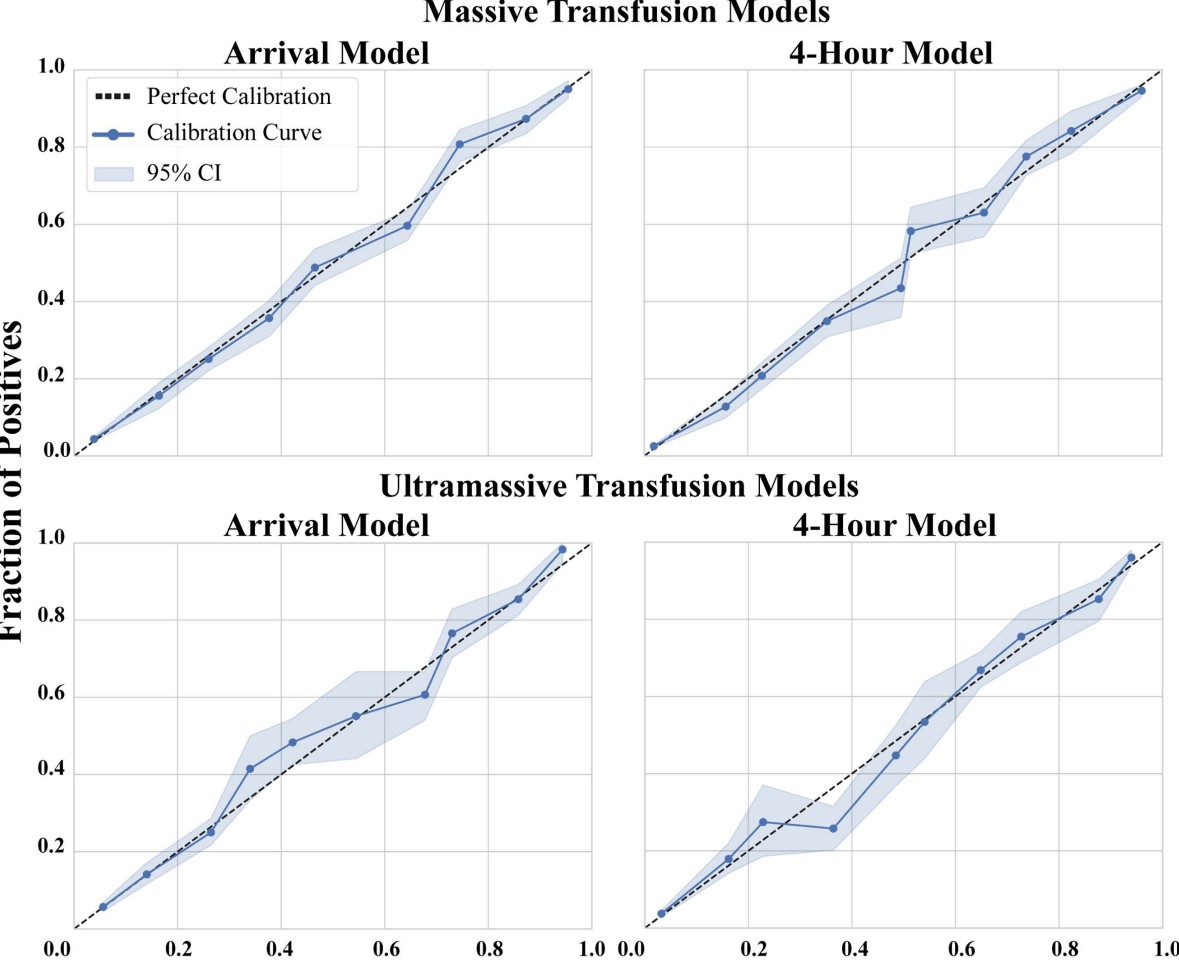

**Fig 2. Reliability diagrams to assess calibration of each model.**

2 model (using variables available on presentation as well as additional vital signs and adjuncts) was able to achieve an AUROC of 0.858 compared to the AUROC of 0.544 using just the ABC score.

Though our model's performance metrics cannot be statistically compared to Benjamin's given the different patient populations and variables in the datasets used in algorithm training, our AUROCs of 0.901 [95% CI 0.895–0.910] and 0.858 [95% CI 0.846–0.872] for our MT and UMT arrival models and 0.943 [95% CI 0.938–0.948]/ 0.922 [95% CI 0.914–0.931] for our 4-hour models suggest that our algorithm performs similarly if not better to existing ML methods. Another strength of our model is that it only uses structured variables that would be available within approximately 60 seconds of patient arrival. This can facilitate direct electronic medical record (EMR) integration where these structured variables could be pulled from the chart and would allow for the model to run directly within the EMR, automatically. Additionally, we specifically chose to train our model on a database that publishes new data every year to counteract model drift and remain stable over time [43].

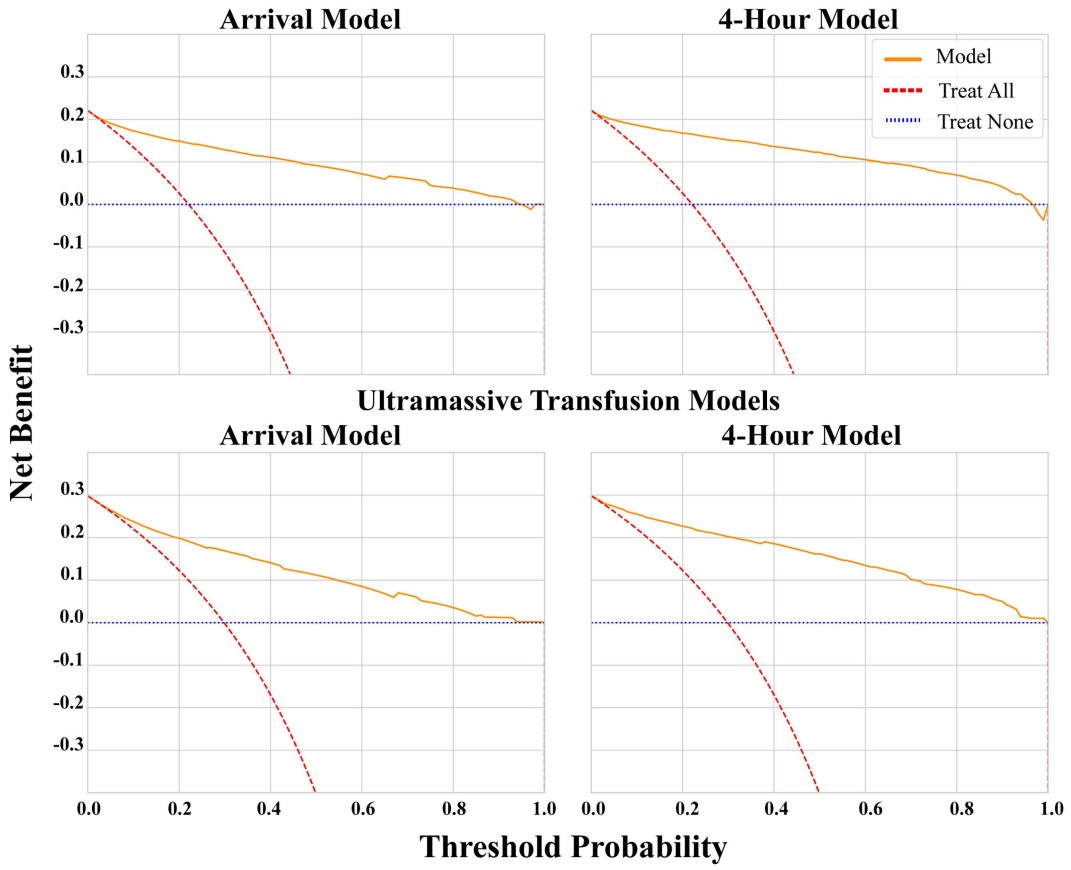

**Fig 3. Decision Curve Analysis to assess net benefit of each model.**

Despite these strengths, our model is not without limitations. Given that these models were trained on a retrospective national database, it lacks the granularity of certain variables that may be important for the prediction of mortality after MT, such as the Focused Assessment with Sonography in Trauma exam, coagulation values, and hemoglobin: all of which were highly impactful features in Benjamin's models for predicting CAT+ status. Because the dataset did not include the components necessary to calculate the ABC or Massive Transfusion Scores, we were unable to benchmark against those established tools. As with all retrospective modeling, the results are only as reliable as the training data. Additionally, though our data was taken from a national database, the model has not yet been prospectively validated, which may limit its generalizability. Our inclusion criterion of patients receiving at least 5 units of blood within the first four hours also introduces a degree of survival bias, as it excludes patients who died before reaching that threshold. Consequently, our model does not apply to patients who did not survive long enough to receive 5 units of blood.

While our model offers a high degree of accuracy in its predictions, it is certainly less interpretable than standard regression-based techniques, an issue common to all ML prediction algorithms. Though we can use ShAP to give us insights into why a given ML model makes the predictions it does, ShAP explanations do not depict independent associations like a regression would. Therefore, interpretation must be done cautiously with the knowledge that these are akin to univariate correlations. For instance, height appears consistently as an important predictor in our model. This does not imply that patient height is

**Table 4. Secondary metrics for each model developed and tested.**

| Model | Threshold | Acc | Sens | Spec | PPV | NPV | F1 | Brier[a] | AUPRC[a] |
|---|---|---|---|---|---|---|---|---|---|
| Massive Transfusion Models | | | | | | | | | |
| Arrival | | | | | | | | | |
| F1 Optimized | 0.29 | 0.856 | 0.761 | 0.883 | 0.648 | 0.929 | 0.700 | 0.0941 | 0.766 |
| 90% Spec | 0.38 | 0.870 | 0.683 | 0.923 | 0.715 | 0.911 | 0.698 | | [0.746-0.785] |
| 4-Hour | | | | | | | | | |
| F1 Optimized | 0.36 | 0.894 | 0.822 | 0.914 | 0.730 | 0.948 | 0.773 | 0.0728 | 0.845 |
| 90% Spec | 0.32 | 0.893 | 0.823 | 0.913 | 0.729 | 0.948 | 0.773 | | [0.829-0.861] |
| Ultramassive Transfusion Models | | | | | | | | | |
| Arrival | | | | | | | | | |
| F1 Optimized | 0.28 | 0.806 | 0.732 | 0.837 | 0.656 | 0.880 | 0.692 | 0.1325 | 0.749 |
| 90% Spec | 0.43 | 0.814 | 0.571 | 0.917 | 0.745 | 0.834 | 0.647 | | [0.726-0.773] |
| 4-Hour | | | | | | | | | |
| F1 Optimized | 0.41 | 0.860 | 0.810 | 0.881 | 0.743 | 0.916 | 0.775 | 0.1002 | 0.847 |
| 90% Spec | 0.50 | 0.864 | 0.751 | 0.911 | 0.783 | 0.896 | 0.767 | | [0.827-0.865] |

Acc: Accuracy; Sens: Sensitivity/recall; specificity; PPV: positive predictive value/precision; NPV: negative predictive value: AUPRC: area under the precision recall curve with 95% Confidence Interval.

[a]These metrics do not depend on threshold and therefore only have one value reported.

associated with mortality, however. It is more likely that height is unlikely to be measured accurately in critical patients and this "missingness" is a proxy for acuity or process-of-care dynamics, rather than a biologically meaningful signal. This phenomenon is consistent with findings from Donohue et al., who showed that missing thromboelastography data correlated with hypotension, low GCS, prehospital intubation, and increased 30-day mortality [44]. As a result, height in these critical patients may be defaulted to a standard value in the EMR, or is left missing and later imputed by our model. It is therefore critical to recognize that this feature analysis is provided to offer transparency into which variables the model is leveraging for its predictions, not to draw causal or explanatory conclusions about the biological or clinical importance of specific variables.

We chose to exclude explicit missingness indicator variables from the models for two reasons. First, to avoid overfitting to institution/database-specific documentation patterns that may not generalize during single-facility implementation, and second, to reduce the risk of the model overly relying on non-physiologic process artifacts. While missingness may carry real prognostic value, it is fragile as a predictor. Future institutional workflows changes, such as mandatory auto-filling of height/weight fields from prior encounters or license records, may result in decreased or entirely absent predictive utility of missingness. As a result, models that lean heavily on these artifacts may fail silently when deployed in environments with different charting practices. Though our model does not explicitly include missingness as a determinant of mortality, it may be able to pick up the imputed values as a sign of this missingness. To address this, separate models were run using random imputation for the height, weight, and temperature variables to assess model performance without this missingness heuristic. While performance of the models did drop slightly on admission (AUROC of 0.870 [95% CI 0.862–0.879] for MT and 0.814 [95% CI 0.801–0.830] for UMT), this drop was less pronounced in the 4-hour models (AUROC of 0.929 [95% CI 0.924–0.935] for MT and 0.904 [95% CI 0.895–0.914] for UMT. Relative feature importance remained similar across all models as well, except for weight and BMI becoming less impactful. Altogether, this suggests that these models are not overly reliant on missing data in their predictions, and that local implementation should take facility-specific practices into account when determining both whether to include explicit missingness and type of imputation method for missingness.

Most importantly though, this predictive tool is not designed to guide the initial activation of MTP or identify patients for whom it should be started. Rather, it is specifically designed to inform decisions about the continuation or termination of

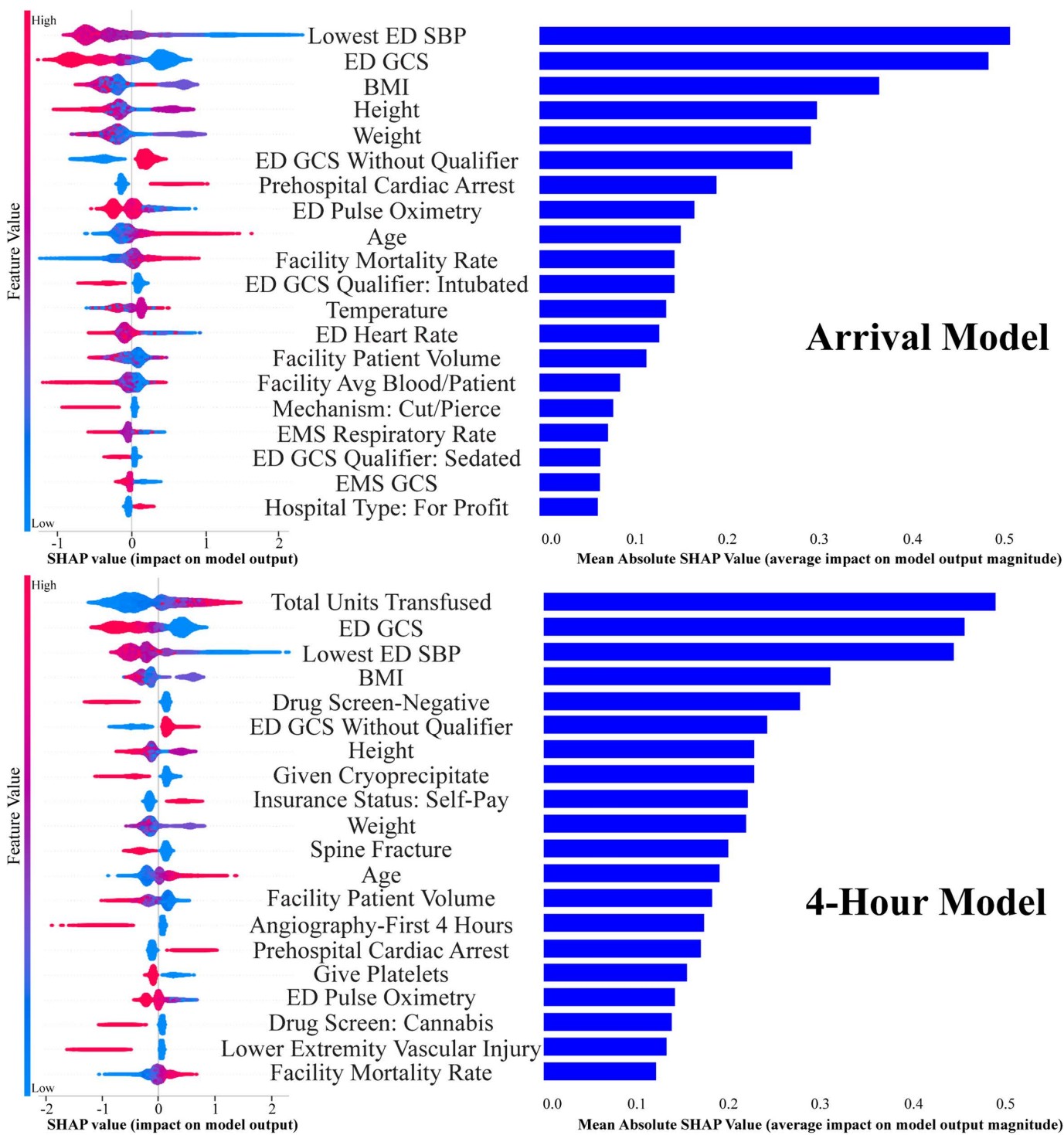

**Fig 4. Shapley additive explanation (SHAP) methods to assess feature importance for both massive transfusion models.** Left side: Beeswarm plot-each point represents an individual patient, the color of each point represents the value of that variable for that patient, and the horizontal displacement represents the effect of that value of that variable on the model's outcome prediction for that individual patient. Right side: Top 20 most important features in relation to the model's decision making, ranked by mean absolute SHAP value. ED: emergency department; SPB: systolic blood pressure; GCS: Glasgow Coma Scale; BMI: body mass index; Avg: average; EMS: emergency medical services.

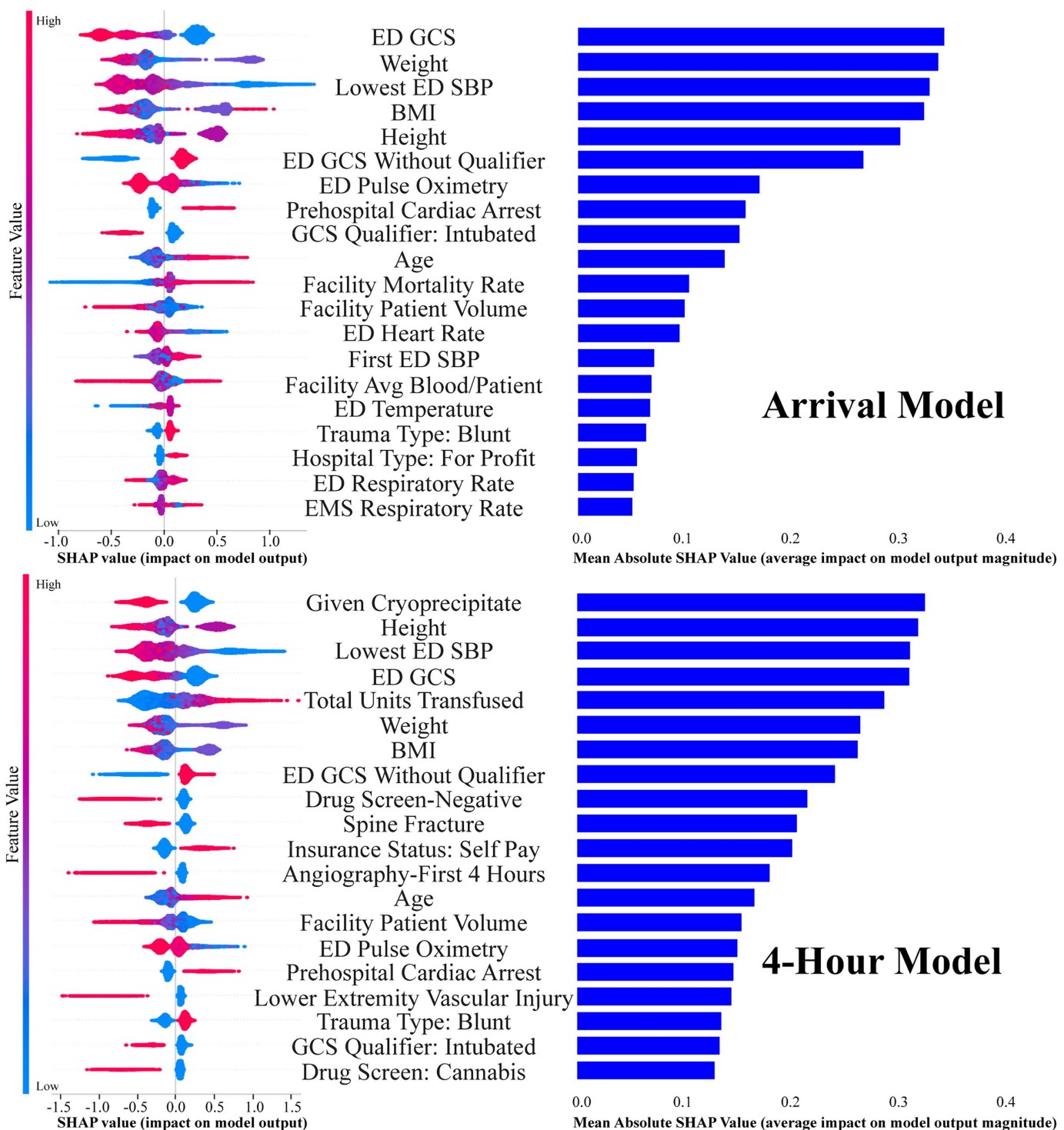

**Fig 5. Shapley additive explanation (SHAP) methods to assess feature importance for all ultramassive transfusion models.** Left side: Beeswarm plot-each point represents an individual patient, the color of each point represents the value of that variable for that patient, and the horizontal displacement represents the effect of that value of that variable on the model's outcome prediction for that individual patient. Right side: Top 20 most important features in relation to the model's decision making, ranked by mean absolute SHAP value. ED: emergency department; GCS: Glasgow Coma Scale; SPB: systolic blood pressure; BMI: body mass index; Avg: average; EMS: emergency medical services.

massive resuscitation once MTP has already been initiated. These are decisions that are currently based largely on clinician gestalt and experience rather than objective, data-driven methods, which may be especially problematic and variable in low-volume, low-acuity centers. Our model aims to supplement, not replace, clinical judgment in this complex setting. No matter how accurate prediction algorithms become, they certainly cannot replace the clinical decision-making of an experienced trauma surgeon. Importantly, while predictive accuracy is high, caution is warranted in interpreting model outputs to avoid a self-fulfilling prophecy, wherein high predicted mortality could lead to premature withdrawal of care. At the same time, the model may offer value in supporting continued resuscitation in patients who might otherwise be deemed futile, particularly in lower-volume trauma centers, where limited experience could lead to underestimation of survivability.

Prospective implementation of this tool must therefore be accompanied by ethical safeguards and institutional calibration. Threshold selection and model integration should reflect each institution's resources, clinical culture, and tolerances for aggressive care. For example, a high predicted mortality score at one institution might trigger an early automated alert to the blood bank signaling imminent MTP activation, while at another, it could serve as an indicator to consider transfer to a higher level of care or an early alert to prepare either the operating room or an ICU bed. Ultimately, the model is intended to enhance, not constrain, individualized clinical decision-making, providing a data-informed method to augment trauma care, especially in resource-limited scenarios.

Our findings demonstrate that ML techniques can predict survival of trauma patients after both MT and UMT with outstanding discrimination and can refine this prediction over time as the resuscitation progresses. The next steps for this model will focus on two distinct areas: prospective validation and model extension using different data modalities. Prospective validation is essential to assess the real-time performance, clinical utility, and integration feasibility of this algorithm across diverse healthcare settings [45]. This will not only confirm the model's predictive accuracy in practice but will also allow for iterative refinement based on institutional-specific workflows, thresholds, and patient populations.

Implementation trials could help evaluate how predictive outputs influence clinician decision-making, and whether they improve resource allocation, patient outcomes, patient communication/prognostication, or the efficiency of MTP continuation decisions. This will also allow for collection of other types of data that can extend the model, such as ED notes, operative reports, radiology reports, and time-series vitals data, as the use of unstructured data like this has been shown to improve predictive accuracy in other ML-based decision support tools [46]. Just as important as technical performance is end-user trust, usability, and buy-in. The successful integration of decision-support tools such as this will depend heavily on how frontline clinicians perceive their relevance, reliability, and ease of use. With this in mind, it will be important to include structured usability assessments such as those informed by the Health Information Technology Usability Assessment Scale framework to ensure that we are developing tools that actually help clinicians at the bedside [47,48]. As the use of ML and AI increases, so will the importance of databases that contain the necessary information for development of these tools. To support the development of robust ML models, future trauma databases should be structured to ensure high-quality, standardized, and clinically relevant variable capture.

## Conclusions

Though ML models have been developed for prediction of mortality in CAT+ patients, this is the first reported ML model to predict 6-hour mortality trauma patients requiring both MT and UMT. A well-calibrated GBDT model performed best with AUROC of 0.901 [95% CI 0.895–0.910] in the arrival model, increasing to 0.943 [95% CI 0.938–0.948] in the 4-hour model (0.858 [95% CI 0.846–0.872] to 0.922 [95% CI 0.914–0.931] in UMT models). This model performs similarly to if not better than previously developed ML models for CAT+ patients. This demonstrates proof-of-concept that ML techniques can be leveraged for development of highly accurate prediction models that may aid clinicians in prognostication as well as making difficult decisions regarding futility and judicious use of blood products in the most critically injured patients.

## Supporting information

**S1 Appendix. PDF of EQUATOR network Transparent Reporting of a multivariable prediction model for Individual Prognosis Or Diagnosis (TRIPOD+AI) guidelines.**
(PDF)

**S2 Appendix. Microsoft word file with a link to github respository containing all code used in training and production of ML models and hyperparameter information.**
(DOCX)

**S3 Appendix. Microsoft excel file containing exhaustive list of Abbreviated-Injury Scale codes used to generate binary variables encoding the presence or absence of injury to various body regions/organ systems.**
(XLSX)

**S4 Appendix. Microsoft excel file containing exhaustive list of all variables used in each specific model described in the manuscript and description of variables.**
(XLSX)

**S5 Appendix. Sensitivity analysis of different missingness thresholds for imputation versus variable removal.**
(PDF)

**S6 Appendix. Structured Technology Summary based on Health Technology Assessment Domains.**
(PDF)

## Author contributions

**Conceptualization:** Michael D. Cobler-Lichter, Nicholas Namias, Brandon M. Parker, Jonathan P. Meizoso.

**Data curation:** Michael D. Cobler-Lichter, Jessica M. Delamater, Brianna L. Collie, Nicole B. Lyons, Luciana Tito Bustillos, Nicholas Namias.

**Formal analysis:** Michael D. Cobler-Lichter.

**Investigation:** Michael D. Cobler-Lichter, Jonathan P. Meizoso.

**Methodology:** Michael D. Cobler-Lichter, Brianna L. Collie, Nicholas Namias, Brandon M. Parker, Jonathan P. Meizoso.

**Resources:** Nicholas Namias, Kenneth G. Proctor.

**Supervision:** Nicholas Namias, Brandon M. Parker, Jonathan P. Meizoso, Kenneth G. Proctor.

**Visualization:** Michael D. Cobler-Lichter.

**Writing – original draft:** Michael D. Cobler-Lichter, Jonathan P. Meizoso, Kenneth G. Proctor.

**Writing – review & editing:** Michael D. Cobler-Lichter, Jessica M. Delamater, Brianna L. Collie, Nicole B. Lyons, Luciana Tito Bustillos, Nicholas Namias, Brandon M. Parker, Jonathan P. Meizoso, Kenneth G. Proctor.

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
