## [Decision Letter · Decision Letter 0]

12 Jul 2025

Dear Dr. Cobler-Lichter,

Thank you for submitting your manuscript to PLOS ONE. After careful consideration, we feel that it has merit but does not fully meet PLOS ONE’s publication criteria as it currently stands. Therefore, we invite you to submit a revised version of the manuscript that addresses the points raised during the review process.

We look forward to receiving your revised manuscript.

Kind regards,

Laila Cure

Academic Editor

PLOS ONE

Journal Requirements:

2. Thank you for stating the following in the Competing Interests/Financial Disclosure section:

I have read the journal's policy and the authors of this manuscript have the following competing interests:

JPM: research support and travel/lodging support from CSL Behring and Takeda Pharmaceuticals; grant support from the U.S. Department of Defense/AMRMC (W81XWH-15-9-0001); speaking honoraria from Cerus Corporation and Memorial Healthcare System; scientific advisory board for Haemonetics, Inc.; editorial board for Trauma Surgery and Acute Care Open and Journal of Surgical Research.

NN: NN receives compensation as an advisory board member for Molnlycke, grand funding from Humacyte, and is on the editorial board of Surgical Infections, and the Journal of Trauma and Acute Care Surgery.

All other authors: no disclosures.

We note that one or more of the authors are employed by a commercial company: CSL Behring, Takeda Pharmaceuticals, Cerus Corporation, Molnlycke, and Humacyte,

3. In the online submission form, you indicated that all code used in development, training, and evaluation of models is provided in the supplementary material. Source data is the property of the Committee on Trauma, American College of Surgeons (NTDB Admission Year 2017-2021, Chicago, IL) but can be provided on request in conjunction with their approval (https://www.facs.org/quality-programs/trauma/quality/national-trauma-data-bank/datasets/ )

4. Please remove all personal information, ensure that the data shared are in accordance with participant consent, and re-upload a fully anonymized data set.

Additional guidance on preparing raw data for publication can be found in our Data Policy (https://journals.plos.org/plosone/s/data-availability#loc-human-research-participant-data-and-other-sensitive-data) and in the following article: http://www.bmj.com/content/340/bmj.c181.long .

Additional Editor Comments:

In addition to the comments from the reviewers, please address the following that will help clarify the value of your machine learning exercise using data from the trauma quality improvement program. I think this manuscript can be improved so that the value of the research is clear to readers with and without trauma decision making background.

1. The introduction should describe how the MT decision is currently made and what criteria emergency physicians use to order MT or UMT. Is this decision made upon arrival to happen 4 hours later? Or once it is decided then it happens almost instantaneously? This will be useful in evaluating your model with respect to the current protocols.

2. You are only including patients that received MT and UMT in your analysis. You are excluding patients that did not receive it. So, your models have no information on what happens to a patient that should have received MT but didn’t (we assume they do not survive?). This also implies that the decision or prediction is related patients who would have already received the MT, so how would you save blood resources if your assumption is that these patients had already received transfusion? Otherwise, how would you know they will receive MT with data upon arrival? This ties to the lack of clarity on the decision implied by the prediction (this should be clear so that we know which predictors and the timing of their measurement are appropriate to include in the model).

3. At the end of your introduction, you are stating that predictions can be “improved further with the incorporation of more clinical information that becomes available as the resuscitation progresses.”, which again, sounds like what physicians already do. However, emergency physicians can explain the logic of their decision making. How would these models (or their insight) fit into the workflow of an emergency care team during resuscitation?

4. When I read the description of the research in the introduction and the methods, it appears that your model is not really for prediction, but for description of the mortality associated with patients already receiving MT and UMT (even though you are using predictive modeling techniques, these can be used for descriptive analyses). You are identifying factors that influence mortality in patients that already received these types of transfusion, most likely following some protocol. If this is the case, then please revise the manuscript to reflect this analysis and avoid the prediction/decision-support confusion.

5. In the model evaluation section, the first sentence should be revised for grammar.

6. In the methods and results section, we need model interpretation for the best performing model in terms of current transfusion protocols. Is there something that your models are suggesting that emergency physicians are not already supposed to look at?

7. How do your models compare to the protocols in practice? Can you tell if the protocol was not followed with the patients based on the dataset?

8. I echo the comments from the reviewers related to the need for being more specific in how the decision-makers would use this information to support their decision. The way the manuscript is written, the decision appears to be whether to withhold transfusion. Please specify who you see as the decision maker. How do you envision this model being used? You mentioned embedded in the medical record (again, your methods imply that this information would appear after the MT or UMT has been implemented).

Reviewers' comments:

Reviewer's Responses to Questions

**Comments to the Author**

1. Is the manuscript technically sound, and do the data support the conclusions?

Reviewer #1: Yes

Reviewer #2: Yes

2. Has the statistical analysis been performed appropriately and rigorously?

Reviewer #1: Yes

Reviewer #2: Yes

3. Have the authors made all data underlying the findings in their manuscript fully available?

Reviewer #1: Yes

Reviewer #2: Yes

4. Is the manuscript presented in an intelligible fashion and written in standard English?

Reviewer #1: Yes

Reviewer #2: Yes

Reviewer #1: This study, utilizing the American College of Surgeons Trauma Quality Improvement Program (TQIP) database, developed a predictive model with high accuracy for 6-hour mortality in trauma patients requiring massive transfusion (MT) or ultra-massive transfusion (UMT). While demonstrating clinical value for assessing post-transfusion risks, the manuscript establishes a prediction model without explicitly providing the predictive formula.

There are some Comments:

1. The authors emphasize the model’s utility in optimizing blood resource allocation and clinical decision-making. However, we wish to highlight a potential ethical consideration: If the model suggests limited therapeutic benefit of transfusion for certain trauma patients, could the authors please discuss how clinicians should balance resource allocation with individualized care? Might withholding transfusion inadvertently exacerbate mortality risks?

2. Please clarify the reasons for selecting the 4-hour and 6-hour timepoints. e.g., from admission time, cumulative transfusion duration, or continuous infusion time and provide a detailed methodological justification for these choices.

3. In the Methods section: The exclusion of variables based on high missing rates may introduce selection bias. Please perform sensitivity analyses to evaluate how varying missingness thresholds impact model robustness.

4. The study does not address potential collinearity among variables, which could affect model stability or parameter interpretation. We suggest adding variance inflation factor (VIF) analyses or similar methods to validate variable independence.

5. In the Results section, weight is identified as a significant predictor. Could the authors please address the practical challenges of obtaining accurate weight measurements in emergency trauma settings?

6. In the Discussion section, please add the mechanistic links between key predictors and transfusion-related outcomes.

Reviewer #2: 1.Thank you for giving me the opportunity to review that manuscript. The given article tackles the highly relevant topic of trauma triage decision-making. The results are certainly relevant to practice and can lead to a change in current practice

The main text of the article is technically complex, especially for readers from the area of clinical care. Therefore, we propose that in the chapter on material and methods, as well as in the secondary objectives, it is structured, as far as possible, in an approximate way based on the following scheme, based on domains that is recommended for Health Technology Assessment, based on domains that range from the description of the health problem to monitoring and post-implementation:

Health Problem and Target Population

A clear description of the health problem and target population is required for intervention.

Definition of the health problem, including prevalence and incidence.

Information on the target population, such as mean age and risk factors.

Details on the standard therapeutic approach.

Technology Description

A detailed description of the technology and its context should be provided.

Main characteristics of the technology and the needs it covers.

Intent to use, including intended and unintended use.

Technical evaluation and validation of the technology.

Technical Aspects of Technology

The technical evaluation focuses on the effectiveness and performance of the technology.

Training data used and their representativeness.

Model evaluation metrics and their clinical justification.

Robustness of the model and its transferability capacity.

Traceability and records management to ensure transparency.

Technology Security

The risks associated with the use of technology are identified and assessed.

Clinical safety and risk assessment for patients and professionals.

Technical security, including privacy and information quality.

Clinical Efficacy and Effectiveness

The clinical benefits of the technology under controlled and uncontrolled conditions are evaluated.

Evidence of effectiveness to support expected benefits.

Analysis of outcome variables according to the intended purpose.

Economic Aspects of Technology

The cost and cost-effectiveness of the technology are analyzed.

Comparison of acquisition, maintenance and use costs.

Evaluation of efficiency and use of resources compared to alternatives.

Of course, it is not necessary to complete all the domains, as some of them would be the subject of one or more subsequent studies.

2.Finally, regarding the discussion and conclusions, we would like to add the following comments that could honestly be the subject of future publications and continuation of such extraordinary work:

Artificial Intelligence (AI) is presented as a tool with enormous potential in the field of medical emergency management due to its characteristics. AI is part of the field of computer science that develops both systems and machines capable of performing tasks that require human intelligence, including Generative Artificial Intelligence and especially LLMs (Large Language Models), that is, that require actions such as learning, reasoning, decision-making, pattern recognition, language comprehension and problem solving. These systems are not governed by strict rules, but are able to learn from data and experiences to improve their performance.

All these applications, such as the one discussed in this study, illustrate how AI, in addition to complementing human capabilities, contributes to improving the effectiveness and efficiency of current management systems in Critical Care. However, the perception of the usefulness and usability of AI in the management of critical medical situations is a fundamental aspect to understand its effectiveness and adoption by responders in emergency situations. Although AI offers great potential, its successful implementation depends largely on how professionals responsible for critical patient care perceive its value and ease of use. In this sense, it is essential to understand in the first instance how users value the capabilities of AI in this context and thus allow the optimization of these systems and guarantee their accessibility and effectiveness.

To this end, three guidelines were proposed on the assessment of usability in this type of tool: visibility, cognitive load and trust. These guidelines were evaluated for the development of the Health Information Technology Usability Assessment Scale (Health-ITUES), a validated instrument that serves as a guide for evaluating technologies applied to health and allows harmonization between different studies. References:

Tan ML, Prasanna R, Stock K, Doyle EEH, Leonard G, Johnston D. Understanding end-users’ perspectives: Towards developing usability guidelines for disaster apps. Progress in Disaster Science. 2020 Oct;7:100118.

Schnall R, Cho H, Liu J. Health Information Technology Usability Evaluation Scale (Health-ITUES) for Usability Assessment of Mobile Health Technology: Validation Study. JMIR Mhealth Uhealth. 2018 Jan 5; 6(1):e4.

**Do you want your identity to be public for this peer review?** For information about this choice, including consent withdrawal, please see our Privacy Policy

Reviewer #1: **Yes: ** Muhan Lü

Reviewer #2: No

---

## [Author Response · Author response to Decision Letter 1]

5 Aug 2025

Additional Editor Comments:

In addition to the comments from the reviewers, please address the following that will help clarify the value of your machine learning exercise using data from the trauma quality improvement program. I think this manuscript can be improved so that the value of the research is clear to readers with and without trauma decision making background.

1. The introduction should describe how the MT decision is currently made and what criteria emergency physicians use to order MT or UMT. Is this decision made upon arrival to happen 4 hours later? Or once it is decided then it happens almost instantaneously? This will be useful in evaluating your model with respect to the current protocols.

We appreciate this insightful comment and agree that clarifying the clinical decision-making process for initiating MTP strengthens the context of our study. In response, we have revised the Introduction (paragraph 2) to include a concise description of how MTP decisions are currently made. Specifically, we now note that MTP activation typically occurs within minutes of patient arrival and is based on limited early data (often just initial vital signs and clinical gestalt) with transfusion beginning almost immediately once activated. We also highlight that decisions regarding ongoing transfusion and prognostication evolve over the following hours and are more complex. This distinction clarifies that our model is designed to support the subsequent phases of care rather than initial MTP activation.

2. You are only including patients that received MT and UMT in your analysis. You are excluding patients that did not receive it. So, your models have no information on what happens to a patient that should have received MT but didn’t (we assume they do not survive?). This also implies that the decision or prediction is related patients who would have already received the MT, so how would you save blood resources if your assumption is that these patients had already received transfusion? Otherwise, how would you know they will receive MT with data upon arrival? This ties to the lack of clarity on the decision implied by the prediction (this should be clear so that we know which predictors and the timing of their measurement are appropriate to include in the model).

We thank the reviewer for this thoughtful comment and apologize for the lack of clarity on our part, which we hope to have corrected with an additional 2 paragraphs early in the discussion (now, paragraphs 2-3 of discussion). This model is not intended to predict which patients should receive MTP, but rather to estimate early mortality among patients who have already begun transfusion. MTP is an ongoing process, and there are multiple points during resuscitation when providers can step back and reassess whether continued transfusion is appropriate. Given the wide range of transfusion volumes delivered after the initial decision for MTP activation (some patients receive 5 units whereas other can receive over 100) our model is designed to support clinical judgment during the evolving resuscitation process by providing individualized risk estimates in real time once the decision to initiate MTP has already begun.

3. At the end of your introduction, you are stating that predictions can be “improved further with the incorporation of more clinical information that becomes available as the resuscitation progresses.”, which again, sounds like what physicians already do. However, emergency physicians can explain the logic of their decision making. How would these models (or their insight) fit into the workflow of an emergency care team during resuscitation?

We fully agree that experienced emergency and trauma physicians already incorporate evolving clinical data into their decision-making during resuscitation and that is precisely what we aim to emulate with this predictive model. Our goal is not to replace clinician judgment, but to support it by quantifying the intuition and pattern recognition that currently guide these decisions. High-stakes choices around ongoing transfusion often rely on gestalt and can vary considerably between providers and institutions. By applying machine learning to large-scale, real-world data, we seek to build a model that mirrors expert reasoning with the added benefit of being evidence-based and data-driven. We fully intend for the physician to remain the ultimate decision-maker; our model is designed to augment that judgment by providing real-time insight at moments when decisions must be made rapidly with limited information. We have clarified this intended role in the revised Discussion (paragraphs 2-3) and added references to support the emphasis of future use of advanced diagnostics, precision medicine, and patient-tailored strategies to inform clinical care in the context of massive transfusion.

4. When I read the description of the research in the introduction and the methods, it appears that your model is not really for prediction, but for description of the mortality associated with patients already receiving MT and UMT (even though you are using predictive modeling techniques, these can be used for descriptive analyses). You are identifying factors that influence mortality in patients that already received these types of transfusion, most likely following some protocol. If this is the case, then please revise the manuscript to reflect this analysis and avoid the prediction/decision-support confusion.

We apologize for the lack of clarity in the original manuscript and appreciate the opportunity to clarify. The primary aim of this study is indeed predictive: once a patient has had MTP initiated, our goal is to estimate the likelihood of 6-hour mortality using information available only at the moment of MTP initiation. However, we acknowledge that resuscitation is not a single event but a dynamic process that unfolds over time. To reflect this, we developed and report two models one using variables available at arrival/MTP initiation, and another incorporating additional data after 4 hours of resuscitation. While we present SHAP-based feature importance analysis, this is not the primary focus of the study. It is included solely to enhance model interpretability and provide clinical insight into how the algorithm weighs different inputs. Notably, we have added cautionary language about how interpretation of this feature analysis should not be done through an explanatory or causal relationship lens, as this type of feature analysis is somewhat akin to a univariate analysis vulnerable to confounding. We agree this component is descriptive in nature and have revised the Methods and Discussion to make clear that the central aim of the study is predictive modeling (i.e. the likelihood of 6-hour mortality in a patient that has had MTP initiated) and performance of that predictive modeling, not identification of risk factors.

5. In the model evaluation section, the first sentence should be revised for grammar.

Thank you for pointing this out, we have corrected this sentence.

6. In the methods and results section, we need model interpretation for the best performing model in terms of current transfusion protocols. Is there something that your models are suggesting that emergency physicians are not already supposed to look at?

We appreciate this important question. Current decisions on continuation of ongoing massive transfusion rely heavily on clinical gestalt which can be inconsistent and subjective, especially outside of high-volume trauma centers. Our model does not suggest that physicians ignore the datapoints they use every day (e.g., GCS, blood pressure, transfusion volume), but instead offers a quantitative, evidence-based tool to formalize and standardize that assessment and provide these clinicians with another piece of quantitative data. Importantly, while experienced trauma teams may intuitively integrate these signals, low-volume or lower-acuity centers may benefit most from predictive support tools that provide real-time risk stratification (consider a situation where a low-volume trauma team judges a patient to be past the point of futility, but the model suggests the patient has a reasonable chance at survival with ongoing resuscitation). By providing individualized mortality predictions based on real data, the model can support more consistent and confident decision-making across a wide range of clinical environments. We have clarified this point in the Discussion (paragraphs 2-3).

7. How do your models compare to the protocols in practice? Can you tell if the protocol was not followed with the patients based on the dataset?

We appreciate this question and the opportunity to clarify. In our study, massive transfusion (MT) and ultramassive transfusion (UMT) are defined by the volume of blood products delivered within a specified time frame (>5 or >10 units within 4 hours, respectively), regardless of whether a formal massive transfusion protocol (MTP) was activated. This volume-based definition is widely used in both clinical and research contexts to identify instances of high-volume resuscitation. While many institutions implement MTPs to standardize delivery (e.g., fixed ratios of red blood cells to platelets to plasma, or predefined sequences of blood product delivery), the TQIP dataset does not capture whether a protocol was formally initiated, adhered to, or deviated from. As such, our models cannot evaluate protocol compliance. Rather, they are designed to predict mortality based on available clinical variables once transfusion is already underway. Importantly, while protocols guide how blood is delivered, there are currently no widely adopted tools or protocols to guide whether continued transfusion is likely to benefit a given patient, an unmet need our model aims to address.

8. I echo the comments from the reviewers related to the need for being more specific in how the decision-makers would use this information to support their decision. The way the manuscript is written, the decision appears to be whether to withhold transfusion. Please specify who you see as the decision maker. How do you envision this model being used? You mentioned embedded in the medical record (again, your methods imply that this information would appear after the MT or UMT has been implemented).

We agree that this should have been more clear. In short, the model is not intended to inform the decision to initiate transfusion or activate MTP, but rather to assist with ongoing decision-making once transfusion has already begun. The intended users are trauma surgeons, emergency physicians, and critical care providers who are managing resuscitation and must regularly reassess the trajectory of care. These decisions often include whether to continue transfusion, escalate care, pursue surgical or interventional options, or consider futility and terminate ongoing resuscitation. We envision this model being integrated into the electronic medical record and triggered at predefined points, such as immediately after the order for MTP activation is placed, automatically after 4 hours, or at specific transfusion thresholds (e.g., 5 or 10 units), to provide real-time mortality risk estimates to inform (but not replace) clinical judgment. We have revised the manuscript (paragraphs 2 and 3 of discussion) to clarify the timing and clinical use case of the model and explicitly identify the decision-makers involved.

Reviewers' comments:

Reviewer's Responses to Questions

Comments to the Author

1. Is the manuscript technically sound, and do the data support the conclusions?

Reviewer #1: Yes

Reviewer #2: Yes

2. Has the statistical analysis been performed appropriately and rigorously?

Reviewer #1: Yes

Reviewer #2: Yes

3. Have the authors made all data underlying the findings in their manuscript fully available?

Reviewer #1: Yes

Reviewer #2: Yes

4. Is the manuscript presented in an intelligible fashion and written in standard English?

Reviewer #1: Yes

Reviewer #2: Yes

5. Review Comments to the Author

Reviewer #1: This study, utilizing the American College of Surgeons Trauma Quality Improvement Program (TQIP) database, developed a predictive model with high accuracy for 6-hour mortality in trauma patients requiring massive transfusion (MT) or ultra-massive transfusion (UMT). While demonstrating clinical value for assessing post-transfusion risks, the manuscript establishes a prediction model without explicitly providing the predictive formula.

There are some Comments:

1. The authors emphasize the model’s utility in optimizing blood resource allocation and clinical decision-making. However, we wish to highlight a potential ethical consideration: If the model suggests limited therapeutic benefit of transfusion for certain trauma patients, could the authors please discuss how clinicians should balance resource allocation with individualized care? Might withholding transfusion inadvertently exacerbate mortality risks?

We absolutely agree that these ethical considerations are paramount in the development and implementation of a model such as this. Any model that predicts limited therapeutic benefit must be used with extreme caution to avoid reinforcing bias or prematurely limiting care. As noted in the manuscript, our model is intended solely to inform physicians in making clinical decisions, and we explicitly caution against using it as a sole determinant for withholding transfusion. Its purpose is to provide additional evidence to support nuanced, individualized care, particularly in complex or resource-limited settings. Regardless of whether clinicians rely on gestalt or predictive models, decisions around ongoing resuscitation are always a delicate balancing act. However, one can also imagine a scenario, particularly in a lower-volume, lower-acuity trauma center, where a patient might be deemed futile by an inexperienced trauma team, yet identified by this model as having a relatively low predicted mortality. In such cases, the model could support continued resuscitation, potentially preventing inappropriate withdrawal of care. We have expanded the Discussion section (paragraphs 10-11) to further address these ethical considerations, including the risk of self-fulfilling prophecy. We emphasize that successful implementation must include institutional safeguards, appropriate threshold selection, and continued reliance on clinical judgment within the broader context of each patient's unique circumstances.

2. Please clarify the reasons for selecting the 4-hour and 6-hour timepoints. e.g., from admission time, cumulative transfusion

---

## [Decision Letter · Decision Letter 1]

28 Sep 2025

Dear Dr. Cobler-Lichter,

Thank you for submitting your manuscript to PLOS ONE. After careful consideration, we feel that it has merit but does not fully meet PLOS ONE’s publication criteria as it currently stands. Therefore, we invite you to submit a revised version of the manuscript that addresses the points raised during the review process.

We look forward to receiving your revised manuscript.

Kind regards,

Laila Cure

Academic Editor

PLOS ONE

Journal Requirements:

Additional Editor Comments:

Page 4 line 87 replace "achieved" with "achieve".

Reviewers' comments:

Reviewer's Responses to Questions

**Comments to the Author**

Reviewer #2: All comments have been addressed

2. Is the manuscript technically sound, and do the data support the conclusions?

Reviewer #2: Yes

3. Has the statistical analysis been performed appropriately and rigorously?

Reviewer #2: Yes

4. Have the authors made all data underlying the findings in their manuscript fully available?

Reviewer #2: Yes

5. Is the manuscript presented in an intelligible fashion and written in standard English?

Reviewer #2: Yes

Reviewer #2: (No Response)

**Do you want your identity to be public for this peer review?** For information about this choice, including consent withdrawal, please see our Privacy Policy

Reviewer #2: No

---

## [Author Response · Author response to Decision Letter 2]

4 Oct 2025

Additional Editor Comments:

Page 4 line 87 replace “achieved” with “achieve”.

We thank the editor for catching this typo and have corrected it.

Reviewer’s Responses to Questions

Comments to the Author

1. If the authors have adequately addressed your comments in a previous round of review and you feel that this manuscript is now acceptable for publication, you may indicate that here to bypass the “Comments to the Author” section, enter your conflict of interest statement in the “Confidential to Editor” section, and submit your “Accept” recommendation.

Reviewer #2: All comments have been addressed

Thank you to all involved reviewers and editors for the time spent reviewing our manuscript and all constructive criticism. We believe it is a much stronger paper as a result.

---

## [Editor Report · Decision Letter 2]

8 Oct 2025

MIASurviveMTP: Machine learning for immediate assessment and survival prediction after Massive Transfusion Protocol

PONE-D-25-20654R2

Dear Dr. Cobler-Lichter,

We’re pleased to inform you that your manuscript has been judged scientifically suitable for publication and will be formally accepted for publication once it meets all outstanding technical requirements.

Kind regards,

Laila Cure

Academic Editor

PLOS ONE
---

## [Editor Report · Acceptance letter]

PONE-D-25-20654R2

PLOS ONE

Dear Dr. Cobler-Lichter,

I'm pleased to inform you that your manuscript has been deemed suitable for publication in PLOS ONE. Congratulations! Your manuscript is now being handed over to our production team.

Kind regards,

on behalf of

Dr. Laila Cure

Academic Editor

PLOS ONE